# Unifying Count-Based Exploration and Intrinsic Motivation

**Marc G. Bellemare**
bellemare@google.com

**Sriram Srinivasan**
srsrinivasan@google.com

**Georg Ostrovski**
ostrovski@google.com

**Tom Schaul**
schaul@google.com

**David Saxton**
saxton@google.com

**Rémi Munos**
munos@google.com

Google DeepMind
London, United Kingdom

## Abstract

We consider an agent's uncertainty about its environment and the problem of generalizing this uncertainty across states. Specifically, we focus on the problem of exploration in non-tabular reinforcement learning. Drawing inspiration from the intrinsic motivation literature, we use density models to measure uncertainty, and propose a novel algorithm for deriving a pseudo-count from an arbitrary density model. This technique enables us to generalize count-based exploration algorithms to the non-tabular case. We apply our ideas to Atari 2600 games, providing sensible pseudo-counts from raw pixels. We transform these pseudo-counts into exploration bonuses and obtain significantly improved exploration in a number of hard games, including the infamously difficult MONTEZUMA'S REVENGE.

## 1  Introduction

Exploration algorithms for Markov Decision Processes (MDPs) are typically concerned with reducing the agent's uncertainty over the environment's reward and transition functions. In a tabular setting, this uncertainty can be quantified using confidence intervals derived from Chernoff bounds, or inferred from a posterior over the environment parameters. In fact, both confidence intervals and posterior shrink as the inverse square root of the state-action visit count $N(x, a)$, making this quantity fundamental to most theoretical results on exploration.

Count-based exploration methods directly use visit counts to guide an agent's behaviour towards reducing uncertainty. For example, Model-based Interval Estimation with Exploration Bonuses (MBIE-EB; Strehl and Littman, 2008) solves the augmented Bellman equation

$$V(x) = \max_{a \in \mathcal{A}} \left[ \hat{R}(x, a) + \gamma \, \mathbb{E}_{\hat{P}} \left[ V(x') \right] + \beta N(x, a)^{-1/2} \right],$$

involving the empirical reward $\hat{R}$, the empirical transition function $\hat{P}$, and an exploration bonus proportional to $N(x, a)^{-1/2}$. This bonus accounts for uncertainties in both transition and reward functions and enables a finite-time bound on the agent's suboptimality.

In spite of their pleasant theoretical guarantees, count-based methods have not played a role in the contemporary successes of reinforcement learning (e.g. Mnih et al., 2015). Instead, most practical methods still rely on simple rules such as $\epsilon$-greedy. The issue is that visit counts are not directly useful in large domains, where states are rarely visited more than once.

Answering a different scientific question, intrinsic motivation aims to provide qualitative guidance for exploration (Schmidhuber, 1991; Oudeyer et al., 2007; Barto, 2013). This guidance can be summarized as "explore what surprises you". A typical approach guides the agent based on change

in prediction error, or *learning progress*. If $e_n(A)$ is the error made by the agent at time $n$ over some event A, and $e_{n+1}(A)$ the same error after observing a new piece of information, then learning progress is

$$e_n(A) - e_{n+1}(A).$$

Intrinsic motivation methods are attractive as they remain applicable in the absence of the Markov property or the lack of a tabular representation, both of which are required by count-based algorithms. Yet the theoretical foundations of intrinsic motivation remain largely absent from the literature, which may explain its slow rate of adoption as a standard approach to exploration.

In this paper we provide formal evidence that intrinsic motivation and count-based exploration are but two sides of the same coin. Specifically, we consider a frequently used measure of learning progress, *information gain* (Cover and Thomas, 1991). Defined as the Kullback-Leibler divergence of a prior distribution from its posterior, information gain can be related to the confidence intervals used in count-based exploration. Our contribution is to propose a new quantity, the *pseudo-count*, which connects information-gain-as-learning-progress and count-based exploration.

We derive our pseudo-count from a density model over the state space. This is in departure from more traditional approaches to intrinsic motivation that consider learning progress with respect to a transition model. We expose the relationship between pseudo-counts, a variant of Schmidhuber's compression progress we call *prediction gain*, and information gain. Combined to Kolter and Ng's negative result on the frequentist suboptimality of Bayesian bonuses, our result highlights the theoretical advantages of pseudo-counts compared to many existing intrinsic motivation methods.

The pseudo-counts we introduce here are best thought of as "function approximation for exploration". We bring them to bear on Atari 2600 games from the Arcade Learning Environment (Bellemare et al., 2013), focusing on games where myopic exploration fails. We extract our pseudo-counts from a simple density model and use them within a variant of MBIE-EB. We apply them to an experience replay setting and to an actor-critic setting, and find improved performance in both cases. Our approach produces dramatic progress on the reputedly most difficult Atari 2600 game, MON-TEZUMA'S REVENGE: within a fraction of the training time, our agent explores a significant portion of the first level and obtains significantly higher scores than previously published agents.

## 2   Notation

We consider a countable state space $\mathcal{X}$. We denote a sequence of length $n$ from $\mathcal{X}$ by $x_{1:n} \in \mathcal{X}^n$, the set of finite sequences from $\mathcal{X}$ by $\mathcal{X}^*$, write $x_{1:n}x$ to mean the concatenation of $x_{1:n}$ and a state $x \in \mathcal{X}$, and denote the empty sequence by $\epsilon$. A *model* over $\mathcal{X}$ is a mapping from $\mathcal{X}^*$ to probability distributions over $\mathcal{X}$. That is, for each $x_{1:n} \in \mathcal{X}^n$ the model provides a probability distribution

$$\rho_n(x) := \rho(x\,;\,x_{1:n}).$$

Note that we do not require $\rho_n(x)$ to be strictly positive for all $x$ and $x_{1:n}$. When it is, however, we may understand $\rho_n(x)$ to be the usual conditional probability of $X_{n+1} = x$ given $X_1 \ldots X_n = x_{1:n}$.

We will take particular interest in the empirical distribution $\mu_n$ derived from the sequence $x_{1:n}$. If $N_n(x) := N(x, x_{1:n})$ is the number of occurrences of a state $x$ in the sequence $x_{1:n}$, then

$$\mu_n(x) := \mu(x\,;\,x_{1:n}) := \frac{N_n(x)}{n}.$$

We call the $N_n$ the *empirical count function*, or simply *empirical count*. The above notation extends to state-action spaces, and we write $N_n(x, a)$ to explicitly refer to the number of occurrences of a state-action pair when the argument requires it. When $x_{1:n}$ is generated by an ergodic Markov chain, for example if we follow a fixed policy in a finite-state MDP, then the limit point of $\mu_n$ is the chain's stationary distribution.

In our setting, a *density model* is any model that assumes states are independently (but not necessarily identically) distributed; a density model is thus a particular kind of generative model. We emphasize that a density model differs from a forward model, which takes into account the temporal relationship between successive states. Note that $\mu_n$ is itself a density model.

# 3 From Densities to Counts

In the introduction we argued that the visit count $N_n(x)$ (and consequently, $N_n(x, a)$) is not directly useful in practical settings, since states are rarely revisited. Specifically, $N_n(x)$ is almost always zero and cannot help answer the question "How novel is this state?" Nor is the problem solved by a Bayesian approach: even variable-alphabet models (e.g. Hutter, 2013) must assign a small, diminishing probability to yet-unseen states. To estimate the uncertainty of an agent's knowledge, we must instead look for a quantity which generalizes across states. Guided by ideas from the intrinsic motivation literature, we now derive such a quantity. We call it a *pseudo-count* as it extends the familiar notion from Bayesian estimation.

## 3.1 Pseudo-Counts and the Recoding Probability

We are given a density model $\rho$ over $\mathcal{X}$. This density model may be approximate, biased, or even inconsistent. We begin by introducing the *recoding probability* of a state $x$:

$$\rho'_n(x) := \rho(x\,;\,x_{1:n}x).$$

This is the probability assigned to $x$ by our density model after observing a new occurrence of $x$. The term "recoding" is inspired from the statistical compression literature, where coding costs are inversely related to probabilities (Cover and Thomas, 1991). When $\rho$ admits a conditional probability distribution,

$$\rho'_n(x) = \mathrm{Pr}_\rho(X_{n+2} = x \mid X_1 \ldots X_n = x_{1:n}, X_{n+1} = x).$$

We now postulate two unknowns: a *pseudo-count function* $\hat{N}_n(x)$, and a *pseudo-count total* $\hat{n}$. We relate these two unknowns through two constraints:

$$\rho_n(x) = \frac{\hat{N}_n(x)}{\hat{n}} \qquad \rho'_n(x) = \frac{\hat{N}_n(x) + 1}{\hat{n} + 1}. \tag{1}$$

In words: we require that, after observing one instance of $x$, the density model's increase in prediction of that same $x$ should correspond to a unit increase in pseudo-count. The pseudo-count itself is derived from solving the linear system (1):

$$\hat{N}_n(x) = \frac{\rho_n(x)(1 - \rho'_n(x))}{\rho'_n(x) - \rho_n(x)} = \hat{n}\rho_n(x). \tag{2}$$

Note that the equations (1) yield $\hat{N}_n(x) = 0$ (with $\hat{n} = \infty$) when $\rho_n(x) = \rho'_n(x) = 0$, and are inconsistent when $\rho_n(x) < \rho'_n(x) = 1$. These cases may arise from poorly behaved density models, but are easily accounted for. From here onwards we will assume a consistent system of equations.

**Definition 1** (Learning-positive density model). *A density model $\rho$ is* learning-positive *if for all $x_{1:n} \in \mathcal{X}^n$ and all $x \in \mathcal{X}$, $\rho'_n(x) \geq \rho_n(x)$.*

By inspecting (2), we see that

1. $\hat{N}_n(x) \geq 0$ if and only if $\rho$ is learning-positive;
2. $\hat{N}_n(x) = 0$ if and only if $\rho_n(x) = 0$; and
3. $\hat{N}_n(x) = \infty$ if and only if $\rho_n(x) = \rho'_n(x)$.

In many cases of interest, the pseudo-count $\hat{N}_n(x)$ matches our intuition. If $\rho_n = \mu_n$ then $\hat{N}_n = N_n$. Similarly, if $\rho_n$ is a Dirichlet estimator then $\hat{N}_n$ recovers the usual notion of pseudo-count. More importantly, if the model generalizes across states then so do pseudo-counts.

## 3.2 Estimating the Frequency of a Salient Event in FREEWAY

As an illustrative example, we employ our method to estimate the number of occurrences of an infrequent event in the Atari 2600 video game FREEWAY (Figure 1, screenshot). We use the Arcade Learning Environment (Bellemare et al., 2013). We will demonstrate the following:

1. Pseudo-counts are roughly zero for novel events,

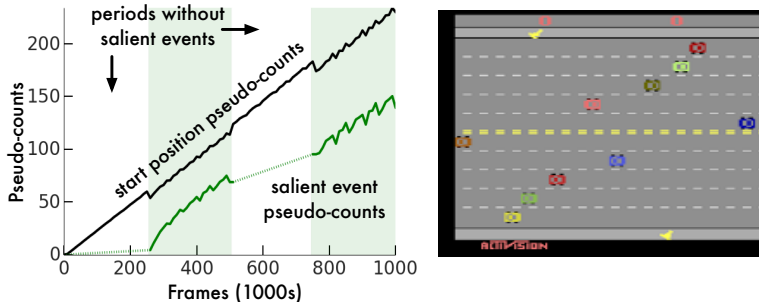

Figure 1: Pseudo-counts obtained from a CTS density model applied to FREEWAY, along with a frame representative of the salient event (crossing the road). Shaded areas depict periods during which the agent observes the salient event, dotted lines interpolate across periods during which the salient event is not observed. The reported values are 10,000-frame averages.

2. they exhibit credible magnitudes,

3. they respect the ordering of state frequency,

4. they grow linearly (on average) with real counts,

5. they are robust in the presence of nonstationary data.

These properties suggest that pseudo-counts provide an appropriate generalized notion of visit counts in non-tabular settings.

In FREEWAY, the agent must navigate a chicken across a busy road. As our example, we consider estimating the number of times the chicken has reached the very top of the screen. As is the case for many Atari 2600 games, this naturally salient event is associated with an increase in score, which ALE translates into a positive reward. We may reasonably imagine that knowing how certain we are about this part of the environment is useful. After crossing, the chicken is teleported back to the bottom of the screen.

To highlight the robustness of our pseudo-count, we consider a nonstationary policy which waits for 250,000 frames, then applies the UP action for 250,000 frames, then waits, then goes UP again. The salient event only occurs during UP periods. It also occurs with the cars in different positions, thus requiring generalization. As a point of reference, we record the pseudo-counts for both the salient event and visits to the chicken's start position.

We use a simplified, pixel-level version of the CTS model for Atari 2600 frames proposed by Bellemare et al. (2014), ignoring temporal dependencies. While the CTS model is rather impoverished in comparison to state-of-the-art density models for images (e.g. Van den Oord et al., 2016), its count-based nature results in extremely fast learning, making it an appealing candidate for exploration. Further details on the model may be found in the appendix.

Examining the pseudo-counts depicted in Figure 1 confirms that they exhibit the desirable properties listed above. In particular, the pseudo-count is almost zero on the first occurrence of the salient event; it increases slightly during the 3rd period, since the salient and reference events share some common structure; throughout, it remains smaller than the reference pseudo-count. The linearity on average and robustness to nonstationarity are immediate from the graph. Note, however, that the pseudo-counts are a fraction of the real visit counts (inasmuch as we can define "real"): by the end of the trial, the start position has been visited about 140,000 times, and the topmost part of the screen, 1285 times. Furthermore, the ratio of recorded pseudo-counts differs from the ratio of real counts. Both effects are quantifiable, as we shall show in Section 5.

## 4    The Connection to Intrinsic Motivation

Having argued that pseudo-counts appropriately generalize visit counts, we will now show that they are closely related to *information gain*, which is commonly used to quantify novelty or curiosity and consequently as an intrinsic reward. Information gain is defined in relation to a *mixture model $\xi$* over

a class of density models $\mathcal{M}$. This model predicts according to a weighted combination from $\mathcal{M}$:

$$\xi_n(x) := \xi(x \, ; \, x_{1:n}) := \int_{\rho \in \mathcal{M}} w_n(\rho)\rho(x \, ; \, x_{1:n})\mathrm{d}\rho,$$

with $w_n(\rho)$ the posterior weight of $\rho$. This posterior is defined recursively, starting from a prior distribution $w_0$ over $\mathcal{M}$:

$$w_{n+1}(\rho) := w_n(\rho, x_{n+1}) \qquad w_n(\rho, x) := \frac{w_n(\rho)\rho(x \, ; \, x_{1:n})}{\xi_n(x)}. \tag{3}$$

Information gain is then the Kullback-Leibler divergence from prior to posterior that results from observing $x$:

$$\mathrm{IG}_n(x) := \mathrm{IG}(x \, ; \, x_{1:n}) := \mathrm{KL}\big(w_n(\cdot, x) \, \| \, w_n\big).$$

Computing the information gain of a complex density model is often impractical, if not downright intractable. However, a quantity which we call the *prediction gain* provides us with a good approximation of the information gain. We define the prediction gain of a density model $\rho$ (and in particular, $\xi$) as the difference between the recoding log-probability and log-probability of $x$:

$$\mathrm{PG}_n(x) := \log \rho'_n(x) - \log \rho_n(x).$$

Prediction gain is nonnegative if and only if $\rho$ is learning-positive. It is related to the pseudo-count:

$$\hat{N}_n(x) \approx \left( e^{\mathrm{PG}_n(x)} - 1 \right)^{-1},$$

with equality when $\rho'_n(x) \to 0$. As the following theorem shows, prediction gain allows us to relate pseudo-count and information gain.

**Theorem 1.** *Consider a sequence $x_{1:n} \in \mathcal{X}^n$. Let $\xi$ be a mixture model over a class of learning-positive models $\mathcal{M}$. Let $\hat{N}_n$ be the pseudo-count derived from $\xi$ (Equation 2). For this model,*

$$IG_n(x) \le PG_n(x) \le \hat{N}_n(x)^{-1} \qquad and \qquad PG_n(x) \le \hat{N}_n(x)^{-1/2}.$$

Theorem 1 suggests that using an exploration bonus proportional to $\hat{N}_n(x)^{-1/2}$, similar to the MBIE-EB bonus, leads to a behaviour at least as exploratory as one derived from an information gain bonus. Since pseudo-counts correspond to empirical counts in the tabular setting, this approach also preserves known theoretical guarantees. In fact, we are confident pseudo-counts may be used to prove similar results in non-tabular settings.

On the other hand, it may be difficult to provide theoretical guarantees about existing bonus-based intrinsic motivation approaches. Kolter and Ng (2009) showed that no algorithm based on a bonus upper bounded by $\beta N_n(x)^{-1}$ for any $\beta > 0$ can guarantee PAC-MDP optimality. Again considering the tabular setting and combining their result to Theorem 1, we conclude that bonuses proportional to immediate information (or prediction) gain are insufficient for theoretically near-optimal exploration: to paraphrase Kolter and Ng, these methods produce explore too little in comparison to pseudo-count bonuses. By inspecting (2) we come to a similar negative conclusion for bonuses proportional to the L1 or L2 distance between $\xi'_n$ and $\xi_n$.

Unlike many intrinsic motivation algorithms, pseudo-counts also do not rely on learning a forward (transition and/or reward) model. This point is especially important because a number of powerful density models for images exist (Van den Oord et al., 2016), and because optimality guarantees cannot in general exist for intrinsic motivation algorithms based on forward models.

## 5  Asymptotic Analysis

In this section we analyze the limiting behaviour of the ratio $\hat{N}_n/N_n$. We use this analysis to assert the consistency of pseudo-counts derived from tabular density models, i.e. models which maintain per-state visit counts. In the appendix we use the same result to bound the approximation error of pseudo-counts derived from directed graphical models, of which our CTS model is a special case.

Consider a fixed, infinite sequence $x_1, x_2, \ldots$ from $\mathcal{X}$. We define the limit of a sequence of functions $\big(f(x \, ; \, x_{1:n}) : n \in \mathbb{N}\big)$ with respect to the length $n$ of the subsequence $x_{1:n}$. We additionally assume that the empirical distribution $\mu_n$ converges pointwise to a distribution $\mu$, and write $\mu'_n(x)$ for the recoding probability of $x$ under $\mu_n$. We begin with two assumptions on our density model.

**Assumption 1.** *The limits*

$$\text{(a) } r(x) := \lim_{n\to\infty} \frac{\rho_n(x)}{\mu_n(x)} \qquad \text{(b) } \dot{r}(x) := \lim_{n\to\infty} \frac{\rho'_n(x) - \rho_n(x)}{\mu'_n(x) - \mu_n(x)}$$

*exist for all $x$; furthermore, $\dot{r}(x) > 0$.*

Assumption (a) states that $\rho$ should eventually assign a probability to $x$ proportional to the limiting empirical distribution $\mu(x)$. In particular there must be a state $x$ for which $r(x) < 1$, unless $\rho_n \to \mu$. Assumption (b), on the other hand, imposes a restriction on the learning rate of $\rho$ relative to $\mu$'s. As both $r(x)$ and $\mu(x)$ exist, Assumption 1 also implies that $\rho_n(x)$ and $\rho'_n(x)$ have a common limit.

**Theorem 2.** *Under Assumption 1, the limit of the ratio of pseudo-counts $\hat{N}_n(x)$ to empirical counts $N_n(x)$ exists for all $x$. This limit is*

$$\lim_{n\to\infty} \frac{\hat{N}_n(x)}{N_n(x)} = \frac{r(x)}{\dot{r}(x)} \left( \frac{1 - \mu(x)r(x)}{1 - \mu(x)} \right).$$

The model's relative rate of change, whose convergence to $\dot{r}(x)$ we require, plays an essential role in the ratio of pseudo- to empirical counts. To see this, consider a sequence $(x_n : n \in \mathbb{N})$ generated i.i.d. from a distribution $\mu$ over a finite state space, and a density model defined from a sequence of nonincreasing step-sizes $(\alpha_n : n \in \mathbb{N})$:

$$\rho_n(x) = (1 - \alpha_n)\rho_{n-1}(x) + \alpha_n \mathbb{I}\{x_n = x\},$$

with initial condition $\rho_0(x) = |\mathcal{X}|^{-1}$. For $\alpha_n = n^{-1}$, this density model is the empirical distribution. For $\alpha_n = n^{-2/3}$, we may appeal to well-known results from stochastic approximation (e.g. Bertsekas and Tsitsiklis, 1996) and find that almost surely

$$\lim_{n\to\infty} \rho_n(x) = \mu(x) \qquad \text{but} \qquad \lim_{n\to\infty} \frac{\rho'_n(x) - \rho_n(x)}{\mu'_n(x) - \mu_n(x)} = \infty.$$

Since $\mu'_n(x) - \mu_n(x) = n^{-1}(1 - \mu'_n(x))$, we may think of Assumption 1(b) as also requiring $\rho$ to converge at a rate of $\Theta(1/n)$ for a comparison with the empirical count $N_n$ to be meaningful. Note, however, that a density model that does not satisfy Assumption 1(b) may still yield useful (but incommensurable) pseudo-counts.

**Corollary 1.** *Let $\phi(x) > 0$ with $\sum_{x\in\mathcal{X}} \phi(x) < \infty$ and consider the count-based estimator*

$$\rho_n(x) = \frac{N_n(x) + \phi(x)}{n + \sum_{x'\in\mathcal{X}} \phi(x')}.$$

*If $\hat{N}_n$ is the pseudo-count corresponding to $\rho_n$ then $\hat{N}_n(x)/N_n(x) \to 1$ for all $x$ with $\mu(x) > 0$.*

# 6 Empirical Evaluation

In this section we demonstrate the use of pseudo-counts to guide exploration. We return to the Arcade Learning Environment, now using the CTS model to generate an exploration bonus.

## 6.1 Exploration in Hard Atari 2600 Games

From 60 games available through the Arcade Learning Environment we selected five "hard" games, in the sense that an $\epsilon$-greedy policy is inefficient at exploring them. We used a bonus of the form

$$R_n^+(x, a) := \beta(\hat{N}_n(x) + 0.01)^{-1/2}, \tag{4}$$

where $\beta = 0.05$ was selected from a coarse parameter sweep. We also compared our method to the optimistic initialization trick proposed by Machado et al. (2015). We trained our agents' Q-functions with Double DQN (van Hasselt et al., 2016), with one important modification: we mixed the Double Q-Learning target with the Monte Carlo return. This modification led to improved results both with and without exploration bonuses (details in the appendix).

Figure 2 depicts the result of our experiment, averaged across 5 trials. Although optimistic initialization helps in FREEWAY, it otherwise yields performance similar to DQN. By contrast, the

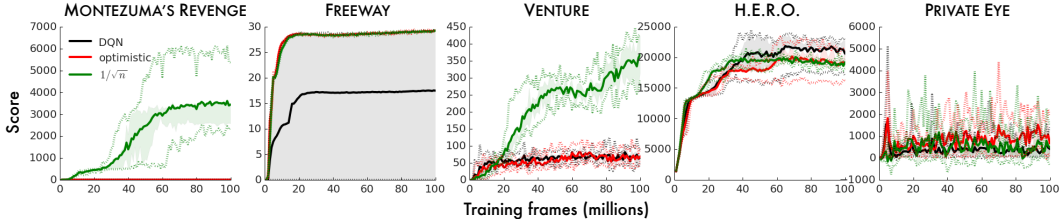

Figure 2: Average training score with and without exploration bonus or optimistic initialization in 5 Atari 2600 games. Shaded areas denote inter-quartile range, dotted lines show min/max scores.

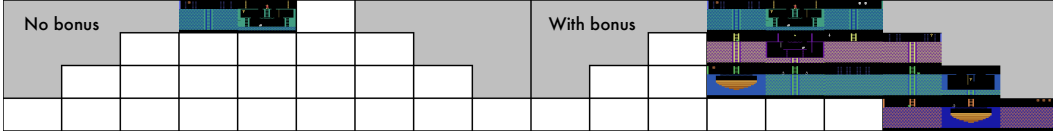

Figure 3: "Known world" of a DQN agent trained for 50 million frames with (**right**) and without (**left**) count-based exploration bonuses, in MONTEZUMA'S REVENGE.

count-based exploration bonus enables us to make quick progress on a number of games, most dramatically in MONTEZUMA'S REVENGE and VENTURE.

MONTEZUMA'S REVENGE is perhaps the hardest Atari 2600 game available through the ALE. The game is infamous for its hostile, unforgiving environment: the agent must navigate a number of different rooms, each filled with traps. Due to its sparse reward function, most published agents achieve an average score close to zero and completely fail to explore most of the 24 rooms that constitute the first level (Figure 3, top). By contrast, within 50 million frames our agent learns a policy which consistently navigates through 15 rooms (Figure 3, bottom). Our agent also achieves a score higher than anything previously reported, with one run consistently achieving 6600 points by 100 million frames (half the training samples used by Mnih et al. (2015)). We believe the success of our method in this game is a strong indicator of the usefulness of pseudo-counts for exploration.[1]

## 6.2 Exploration for Actor-Critic Methods

We next used our exploration bonuses in conjunction with the A3C (Asynchronous Advantage Actor-Critic) algorithm of Mnih et al. (2016). One appeal of actor-critic methods is their explicit separation of policy and Q-function parameters, which leads to a richer behaviour space. This very separation, however, often leads to deficient exploration: to produce any sensible results, the A3C policy must be regularized with an entropy cost. We trained A3C on 60 Atari 2600 games, with and without the exploration bonus (4). We refer to our augmented algorithm as A3C+. Full details and additional results may be found in the appendix.

We found that A3C fails to learn in **15** games, in the sense that the agent does not achieve a score 50% better than random. In comparison, there are only **10** games for which A3C+ fails to improve on the random agent; of these, **8** are games where DQN fails in the same sense. We normalized the two algorithms' scores so that 0 and 1 are respectively the minimum and maximum of the random agent's and A3C's end-of-training score on a particular game. Figure 4 depicts the in-training median score for A3C and A3C+, along with 1st and 3rd quartile intervals. Not only does A3C+ achieve slightly superior median performance, but it also significantly outperforms A3C on at least a quarter of the games. This is particularly important given the large proportion of Atari 2600 games for which an $\epsilon$-greedy policy is sufficient for exploration.

## 7 Related Work

Information-theoretic quantities have been repeatedly used to describe intrinsically motivated behaviour. Closely related to prediction gain is Schmidhuber (1991)'s notion of compression progress,

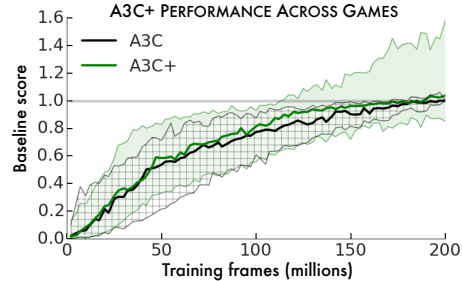

Figure 4: Median and interquartile performance across 60 Atari 2600 games for A3C and A3C+.

which equates novelty with an agent's improvement in its ability to compress its past. More recently, Lopes et al. (2012) showed the relationship between time-averaged prediction gain and visit counts in a tabular setting; their result is a special case of Theorem 2. Orseau et al. (2013) demonstrated that maximizing the sum of future information gains does lead to optimal behaviour, even though maximizing immediate information gain does not (Section 4). Finally, there may be a connection between sequential normalized maximum likelihood estimators and our pseudo-count derivation (see e.g. Ollivier, 2015).

Intrinsic motivation has also been studied in reinforcement learning proper, in particular in the context of discovering skills (Singh et al., 2004; Barto, 2013). Recently, Stadie et al. (2015) used a squared prediction error bonus for exploring in Atari 2600 games. Closest to our work is Houthooft et al. (2016)'s variational approach to intrinsic motivation, which is equivalent to a second order Taylor approximation to prediction gain. Mohamed and Rezende (2015) also considered a variational approach to the different problem of maximizing an agent's ability to influence its environment.

Aside for Orseau et al.'s above-cited work, it is only recently that theoretical guarantees for exploration have emerged for non-tabular, stateful settings. We note Pazis and Parr (2016)'s PAC-MDP result for metric spaces and Leike et al. (2016)'s asymptotic analysis of Thompson sampling in general environments.

## 8 Future Directions

The last few years have seen tremendous advances in learning representations for reinforcement learning. Surprisingly, these advances have yet to carry over to the problem of exploration. In this paper, we reconciled counts, the fundamental unit of uncertainty, with prediction-based heuristics and intrinsic motivation. Combining our work with more ideas from deep learning and better density models seems a plausible avenue for quick progress in practical, efficient exploration. We now conclude by outlining a few research directions we believe are promising.

**Induced metric.** We did not address the question of *where* the generalization comes from. Clearly, the choice of density model induces a particular metric over the state space. A better understanding of this metric should allow us to tailor the density model to the problem of exploration.

**Compatible value function.** There may be a mismatch in the learning rates of the density model and the value function: DQN learns much more slowly than our CTS model. As such, it should be beneficial to design value functions compatible with density models (or vice-versa).

**The continuous case.** Although we focused here on countable state spaces, we can as easily define a pseudo-count in terms of probability density functions. At present it is unclear whether this provides us with the right notion of counts for continuous spaces.

### Acknowledgments

The authors would like to thank Laurent Orseau, Alex Graves, Joel Veness, Charles Blundell, Shakir Mohamed, Ivo Danihelka, Ian Osband, Matt Hoffman, Greg Wayne, Will Dabney, and Aäron van den Oord for their excellent feedback early and late in the writing, and Pierre-Yves Oudeyer and Yann Ollivier for pointing out additional connections to the literature.

## Footnotes

[1] A video of our agent playing is available at `https://youtu.be/0yI2wJ6F8r0`.

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
