[Supplementary Material · pseudo-counts-supplemental.pdf]

# Unifying Count-Based Exploration and Intrinsic Motivation
# Paper and Supplemental

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

[2]Technically, the ALE is partially observable and a frame is an observation, not a state. In many games, however, the current frame is sufficiently informative to guide exploration.

[3]We emphasize that the game RAM is not made available to the agent, and is solely used here in our behavioural analysis.

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

## A  The Connection to Intrinsic Motivation

The following provides an identity connecting information gain and prediction gain.

**Lemma 1.** *Consider a mixture model $\xi$ over $\mathcal{M}$ with prediction gain $PG_n$ and information gain $IG_n$, a fixed $x \in \mathcal{X}$, and let $w'_n(x) := w_n(\rho, x)$ be the posterior of $\xi$ over $\mathcal{M}$ after observing $x$. Let $w''_n(x) := w'_n(\rho, x)$ be the same posterior after observing $x$ a second time, and let $PG_n^\rho(x)$ denote the prediction gain of $\rho \in \mathcal{M}$. Then*

$$PG_n(x) = KL(w'_n \,\|\, w_n) + KL(w'_n \,\|\, w''_n) = IG_n(x) + KL(w'_n \,\|\, w''_n) + \mathbb{E}_{w'_n}\left[PG_n^\rho(x)\right].$$

*In particular, if $\mathcal{M}$ is a class of non-adaptive models in the sense that $\rho_n(x) = \rho(x)$ for all $x_{1:n}$, then*

$$PG_n(x) = KL(w'_n \,\|\, w_n) + KL(w'_n \,\|\, w''_n) = IG_n(x) + KL(w'_n \,\|\, w''_n).$$

A model which is non-adaptive is also learning-positive in the sense of Definition 1. Many common mixture models, for example Dirichlet-multinomial estimators, are mixtures over non-adaptive models.

*Proof.* We rewrite the posterior update rule (3) to show that for any $\rho \in \mathcal{M}$ and any $x \in \mathcal{X}$,

$$\xi_n(x) = \frac{\rho_n(x) w_n(\rho)}{w_n(\rho, x)}.$$

Write $\mathbb{E}_{w'_n} := \mathbb{E}_{\rho \sim w'_n(\cdot)}$. Now

$$
\begin{aligned}
PG_n(x) = \log \frac{\xi'_n(x)}{\xi_n(x)} &= \mathbb{E}_{w'_n}\left[\log \frac{\xi'_n(x)}{\xi_n(x)}\right] \\
&= \mathbb{E}_{w'_n}\left[\log \frac{w'_n(\rho)}{w''_n(\rho)} \frac{w'_n(\rho)}{w_n(\rho)} \frac{\rho'_n(x)}{\rho_n(x)}\right] \\
&= \mathbb{E}_{w'_n}\left[\log \frac{w'_n(\rho)}{w_n(\rho)}\right] + \mathbb{E}_{w'_n}\left[\log \frac{w'_n(\rho)}{w''_n(\rho)}\right] + \mathbb{E}_{w'_n}\left[\log \frac{\rho'_n(x)}{\rho_n(x)}\right] \\
&= IG_n(x) + KL(w'_n \,\|\, w''_n) + \mathbb{E}_{w'_n}\left[PG_n^\rho(x)\right]. \qquad \square
\end{aligned}
$$

The second statement follows immediately.

**Lemma 2.** *The functions $f(x) := e^x - 1 - x$ and $g(x) := e^x - 1 - x^2$ are nonnegative on $x \in [0, \infty)$.*

*Proof.* The statement regarding $f(x)$ follows directly from the Taylor expansion for $e^x$. Now, the first derivative of $g(x)$ is $e^x - 2x$. It is clearly positive for $x \geq 1$. For $x \in [0, 1]$,

$$e^x - 2x = \sum_{i=0}^{\infty} \frac{x^i}{i!} - 2x \geq 1 - x \geq 0.$$

Since $g(0) = 0$, the second result follows. $\qquad \square$

*Proof (Theorem 1).* The inequality $\mathrm{IG}_n(x) \le \mathrm{PG}_n(x)$ follows directly from Lemma 1, the nonnegativity of the Kullback-Leibler divergence, and the fact that all models in $\mathcal{M}$ are learning-positive. For the inequality $\mathrm{PG}_n(x) \le \hat{N}_n(x)^{-1}$, we write

$$\hat{N}_n(x)^{-1} = (1 - \xi'_n(x))^{-1} \frac{\xi'_n(x) - \xi_n(x)}{\xi_n(x)}$$

$$= (1 - \xi'_n(x))^{-1} \left( \frac{\xi'_n(x)}{\xi_n(x)} - 1 \right)$$

$$\overset{(a)}{=} (1 - \xi'_n(x))^{-1} \left( e^{\mathrm{PG}_n(x)} - 1 \right)$$

$$\overset{(b)}{\ge} e^{\mathrm{PG}_n(x)} - 1$$

$$\overset{(c)}{\ge} \mathrm{PG}_n(x),$$

where (a) follows by definition of prediction gain, (b) from $\xi'_n(x) \in [0, 1)$, and (c) from Lemma 2. Using the second part of Lemma 2 in (c) yields the inequality $\hat{N}_n(x)^{-1/2} \ge \mathrm{PG}_n(x)$. $\qquad\square$

## B   Asymptotic Analysis

We begin with a simple lemma which will prove useful throughout.

**Lemma 3.** *The rate of change of the empirical distribution, $\mu'_n(x) - \mu_n(x)$, is such that*

$$n\big(\mu'_n(x) - \mu_n(x)\big) = 1 - \mu'_n(x).$$

*Proof.* We expand the definition of $\mu_n$ and $\mu'_n$:

$$n\big(\mu'_n(x) - \mu_n(x)\big) = n \left[ \frac{N_n(x) + 1}{n+1} - \frac{N_n(x)}{n} \right]$$

$$= \left[ \frac{n}{n+1} \big(N_n(x) + 1\big) - N_n(x) \right]$$

$$= \left[ 1 - \frac{N_n(x) + 1}{n+1} \right]$$

$$= 1 - \mu'_n(x).$$

$\qquad\square$

Using this lemma, we derive an asymptotic relationship between $N_n$ and $\hat{N}_n$.

*Proof (Theorem 2).* We expand the definition of $\hat{N}_n(x)$ and $N_n(x)$:

$$\frac{\hat{N}_n(x)}{N_n(x)} = \frac{\rho_n(x)(1 - \rho'_n(x))}{N_n(x)(\rho'_n(x) - \rho_n(x))}$$

$$= \frac{\rho_n(x)(1 - \rho'_n(x))}{n\mu_n(x)(\rho'_n(x) - \rho_n(x))}$$

$$= \frac{\rho_n(x)(\mu'_n(x) - \mu_n(x))}{\mu_n(x)(\rho'_n(x) - \rho_n(x))} \frac{1 - \rho'_n(x)}{n(\mu'_n(x) - \mu_n(x))}$$

$$= \frac{\rho_n(x)}{\mu_n(x)} \frac{\mu'_n(x) - \mu_n(x)}{\rho'_n(x) - \rho_n(x)} \frac{1 - \rho'_n(x)}{1 - \mu'_n(x)},$$

with the last line following from Lemma 3. Under Assumption 1, all terms of the right-hand side converge as $n \to \infty$. Taking the limit on both sides,

$$\lim_{n \to \infty} \frac{\hat{N}_n(x)}{N_n(x)} \overset{(a)}{=} \frac{r(x)}{\dot{r}(x)} \lim_{n \to \infty} \frac{1 - \rho'_n(x)}{1 - \mu'_n(x)}$$

$$\overset{(b)}{=} \frac{r(x)}{\dot{r}(x)} \frac{1 - \mu(x)r(x)}{1 - \mu(x)},$$

where (a) is justified by the existence of the relevant limits and $\dot{r}(x) > 0$, and (b) follows from writing $\rho'_n(x)$ as $\mu_n(x)\rho'_n(x)/\mu_n(x)$, where all limits involved exist. $\qquad\square$

## B.1   Directed Graphical Models

We say that $\mathcal{X}$ is a *factored* state space if it is the Cartesian product of $k$ subspaces, i.e. $\mathcal{X} := \mathcal{X}_1 \times \cdots \times \mathcal{X}_k$. This factored structure allows us to construct approximate density models over $\mathcal{X}$, for example by modelling the joint density as a product of marginals. We write the $i^{th}$ factor of a state $x \in \mathcal{X}$ as $x^i$, and write the sequence of the $i^{th}$ factor across $x_{1:n}$ as $x^i_{1:n}$.

We will show that directed graphical models (Wainwright and Jordan, 2008) satisfy Assumption 1. A directed graphical model describes a probability distribution over a factored state space. To the $i^{th}$ factor $x^i$ is associated a parent set $\pi(i) \subseteq \{1, \dots, i-1\}$. Let $x^{\pi(i)}$ denote the value of the factors in the parent set. The $i^{th}$ factor model is $\rho^i_n(x^i\,;\,x^{\pi(i)}) := \rho^i(x^i\,;\,x_{1:n}, x^{\pi(i)})$, with the understanding that $\rho^i$ is allowed to make a different prediction for each value of $x^{\pi(i)}$. The state $x$ is assigned the joint probability

$$\rho_{\text{GM}}(x\,;\,x_{1:n}) := \prod_{i=1}^{k} \rho^i_n(x^i\,;\,x^{\pi(i)}).$$

Common choices for $\rho^i_n$ include the conditional empirical distribution and the Dirichlet estimator.

**Proposition 1.** *Suppose that each factor model $\rho^i_n$ converges to the conditional probability distribution $\mu(x^i \mid x^{\pi(i)})$ and that for each $x^i$ with $\mu(x^i \mid x^{\pi(i)})$,*

$$\lim_{n\to\infty} \frac{\rho^i(x^i\,;\,x_{1:n}x, x^{\pi(i)}) - \rho^i(x^i\,;\,x_{1:n}, x^{\pi(i)})}{\mu(x^i\,;\,x_{1:n}x, x^{\pi(i)}) - \mu(x^i\,;\,x_{1:n}, x^{\pi(i)})} = 1.$$

*Then for all $x$ with $\mu(x) > 0$, the density model $\rho_{\text{GM}}$ satisfies Assumption 1 with*

$$r(x) = \frac{\prod_{i=1}^{k} \mu(x^i \mid x^{\pi(i)})}{\mu(x)} \qquad \text{and} \qquad \dot{r}(x) = \frac{\sum_{i=1}^{k}\left(1 - \mu(x^i \mid x^{\pi(i)})\right)\prod_{j\neq i}\mu(x^j \mid x^{\pi(j)})}{1 - \mu(x)}.$$

The CTS density model used in our experiments is in fact a particular kind of induced graphical model. The result above thus describes how the pseudo-counts computed in Section 3.2 are asymptotically related to the empirical counts.

*Proof.* By hypothesis, $\rho^i_n \to \mu(x^i \mid x^{\pi(i)})$. Combining this with $\mu_n(x) \to \mu(x) > 0$,

$$\begin{aligned}
r(x) &= \lim_{n\to\infty} \frac{\rho_{\text{DGM}}(x\,;\,x_{1:n})}{\mu_n(x)} \\
&= \lim_{n\to\infty} \frac{\prod_{i=1}^{k}\rho^i_n(x^i\,;\,x^{\pi(i)})}{\mu_n(x)} \\
&= \frac{\prod_{i=1}^{k}\mu(x^i \mid x^{\pi(i)})}{\mu(x)}.
\end{aligned}$$

Similarly,

$$\begin{aligned}
\dot{r}(x) &= \lim_{n\to\infty} \frac{\rho'_{\text{DGM}}(x\,;\,x_{1:n}) - \rho_{\text{DGM}}(x\,;\,x_{1:n})}{\mu'_n(x) - \mu_n(x)} \\
&\overset{(a)}{=} \lim_{n\to\infty} \frac{\left(\rho'_{\text{DGM}}(x\,;\,x_{1:n}) - \rho_{\text{DGM}}(x\,;\,x_{1:n})\right)n}{1 - \mu'_n(x)} \\
&= \lim_{n\to\infty} \frac{\left(\rho'_{\text{DGM}}(x\,;\,x_{1:n}) - \rho_{\text{DGM}}(x\,;\,x_{1:n})\right)n}{1 - \mu(x)},
\end{aligned}$$

where in (a) we used the identity $n(\mu'_n(x) - \mu_n(x)) = 1 - \mu'_n(x)$ derived in the proof of Theorem 2. Now

$$\dot{r}(x) = (1 - \mu(x))^{-1} \lim_{n \to \infty} \left( \rho'_{\text{DGM}}(x \,;\, x_{1:n}) - \rho_{\text{DGM}}(x \,;\, x_{1:n}) \right) n$$

$$= (1 - \mu(x))^{-1} \lim_{n \to \infty} \left( \prod_{i=1}^{k} \rho^i(x^i \,;\, x_{1:n}x, x^{\pi(i)}) - \prod_{i=1}^{k} \rho^i(x^i \,;\, x_{1:n}, x^{\pi(i)}) \right) n.$$

Let $c_i := \rho^i(x^i \,;\, x_{1:n}, x^{\pi(i)})$ and $c'_i := \rho^i(x^i \,;\, x_{1:n}x, x^{\pi(i)})$. The difference of products above is

$$\left( \prod_{i=1}^{k} \rho^i(x^i \,;\, x_{1:n}x, x^{\pi(i)}) - \prod_{i=1}^{k} \rho^i(x^i \,;\, x_{1:n}, x^{\pi(i)}) \right) = (c'_1 c'_2 \dots c'_k - c_1 c_2 \dots c_k)$$

$$= (c'_1 - c_1)(c'_2 \dots c'_k) + c_1(c'_2 \dots c'_k - c_2 \dots c_k)$$

$$= \sum_{i=1}^{k} (c'_i - c_i) \left( \prod_{j<i} c_j \right) \left( \prod_{j>i} c'_j \right),$$

and

$$\dot{r}(x) = (1 - \mu(x))^{-1} \lim_{n \to \infty} \sum_{i=1}^{k} n(c'_i - c_i) \left( \prod_{j<i} c_j \right) \left( \prod_{j>i} c'_j \right).$$

By the hypothesis on the rate of change of $\rho^i$ and the identity $n \left( \mu(x^i \,;\, x_{1:n}x, x^{\pi(i)}) - \mu(x^i \,;\, x_{1:n}, x^{\pi(i)}) \right) = 1 - \mu(x^i \mid x^{\pi(i)})$, we have

$$\lim_{n \to \infty} n(c'_i - c_i) = 1 - \mu(x^i \mid x^{\pi(i)}).$$

Since the limits of $c'_i$ and $c_i$ are both $\mu(x^i \mid x^{\pi(i)})$, we deduce that

$$\dot{r}(x) = \frac{\sum_{i=1}^{k} \left( 1 - \mu(x^i \mid x^{\pi(i)}) \prod_{j \neq i} \mu(x^j \mid x^{\pi_j(x)}) \right)}{1 - \mu(x)}.$$

Now, if $\mu(x) > 0$ then also $\mu(x^i \,;\, x^{\pi(i)}) > 0$ for each factor $x^i$. Hence $\dot{r}(x) > 0$. □

## B.2 Tabular Density Models (Corollary 1)

We shall prove the following, which includes Corollary 1 as a special case.

**Lemma 4.** *Consider $\phi : \mathcal{X} \times \mathcal{X}^* \to \mathbb{R}^+$. Suppose that for all $(x_n : n \in \mathbb{N})$ and every $x \in \mathcal{X}$*

*1. $\lim\limits_{n \to \infty} \frac{1}{n} \sum\limits_{x \in \mathcal{X}} \phi(x, x_{1:n}) = 0$, and*

*2. $\lim\limits_{n \to \infty} \left( \phi(x, x_{1:n}x) - \phi(x, x_{1:n}) \right) = 0$.*

*Let $\rho_n(x)$ be the count-based estimator*

$$\rho_n(x) = \frac{N_n(x) + \phi(x, x_{1:n})}{n + \sum_{x \in \mathcal{X}} \phi(x, x_{1:n})}.$$

*If $\hat{N}_n$ is the pseudo-count corresponding to $\rho_n$ then $\hat{N}_n(x)/N_n(x) \to 1$ for all $x$ with $\mu(x) > 0$.*

Condition 2 is satisfied if $\phi_n(x, x_{1:n}) = u_n(x)\phi_n$ with $\phi_n$ monotonically increasing in $n$ (but not too quickly!) and $u_n(x)$ converging to some distribution $u(x)$ for all sequences $(x_n : n \in \mathbb{N})$. This is the case for most tabular density models.

*Proof.* We will show that the condition on the rate of change required by Proposition 1 is satisfied under the stated conditions. Let $\phi_n(x) := \phi(x, x_{1:n})$, $\phi'_n(x) := \phi(x, x_{1:n}x)$, $\phi_n := \sum_{x \in \mathcal{X}} \phi_n(x)$ and $\phi'_n := \sum_{x \in \mathcal{X}} \phi'_n(x)$. By hypothesis,

$$\rho_n(x) = \frac{N_n(x) + \phi_n(x)}{n + \phi_n} \qquad \rho'_n(x) = \frac{N_n(x) + \phi'_n(x) + 1}{n + \phi'_n + 1}.$$

Note that we do not require $\phi_n(x) = \phi'_n(x)$. Now

$$
\begin{aligned}
\rho'_n(x) - \rho_n(x) &= \frac{n + \phi_n}{n + \phi_n} \rho'_n(x) - \rho_n(x) \\
&= \frac{n + 1 + \phi'_n}{n + \phi_n} \rho'_n(x) - \rho_n(x) - \frac{(1 + (\phi'_n - \phi_n))\rho'_n(x)}{n + \phi_n} \\
&= \frac{1}{n + \phi_n} \Big[ (N_n(x) + 1 + \phi'_n(x) - (N_n(x) + \phi_n(x))) - (1 + (\phi'_n - \phi_n))\rho'_n(x) \Big] \\
&= \frac{1}{n + \phi_n} \Big[ 1 - \rho'_n(x) + \big(\phi'_n(x) - \phi_n(x)\big) - \rho'_n(x)\big(\phi'_n - \phi_n\big) \Big].
\end{aligned}
$$

Using Lemma 3 we deduce that

$$
\frac{\rho'_n(x) - \rho_n(x)}{\mu'_n(x) - \mu_n(x)} = \frac{n}{n + \phi_n} \frac{1 - \rho'_n(x) + \phi'_n(x) - \phi_n(x) + \rho'_n(x)(\phi'_n - \phi_n)}{1 - \mu'_n(x)}.
$$

Since $\phi_n = \sum_x \phi_n(x)$ and similarly for $\phi'_n$, then $\phi'_n(x) - \phi_n(x) \to 0$ pointwise implies that $\phi'_n - \phi_n \to 0$ also. For any $\mu(x) > 0$,

$$
\begin{aligned}
0 \leq \lim_{n \to \infty} \frac{\phi_n(x)}{N_n(x)} &\overset{(a)}{\leq} \lim_{n \to \infty} \frac{\sum_{x \in \mathcal{X}} \phi_n(x)}{N_n(x)} \\
&= \lim_{n \to \infty} \frac{\sum_{x \in \mathcal{X}} \phi_n(x)}{n} \frac{n}{N_n(x)} \\
&\overset{(b)}{=} 0,
\end{aligned}
$$

where a) follows from $\phi_n(x) \geq 0$ and b) is justified by $n/N_n(x) \to \mu(x)^{-1} > 0$ and the hypothesis that $\sum_{x \in \mathcal{X}} \phi_n(x)/n \to 0$. Therefore $\rho_n(x) \to \mu(x)$. Hence

$$
\lim_{n \to \infty} \frac{\rho'_n(x) - \rho_n(x)}{\mu'_n(x) - \mu_n(x)} = \lim_{n \to \infty} \frac{n}{n + \phi_n} \frac{1 - \rho'_n(x)}{1 - \mu'_n(x)} = 1.
$$

Since $\rho_n(x) \to \mu(x)$, we further deduce from Theorem 2 that

$$
\lim_{n \to \infty} \frac{\hat{N}_n(x)}{N_n(x)} = 1. \qquad \qquad \square
$$

The condition $\mu(x) > 0$, which was also needed in Proposition 1, is necessary for the ratio to converge to 1: for example, if $N_n(x)$ grows as $O(\log n)$ but $\phi_n(x)$ grows as $O(\sqrt{n})$ (with $|\mathcal{X}|$ finite) then $\hat{N}_n(x)$ will grow as the larger $\sqrt{n}$.

## C  Experimental Methods

### C.1  CTS Density Model

Our state space $\mathcal{X}$ is the set of all preprocessed Atari 2600 frames.[2] Each raw frame is composed of $210 \times 160$ 7-bit NTSC pixels (Bellemare et al., 2013). We preprocess these frames by first converting them to grayscale (luminance), then downsampling to $42 \times 42$ by averaging over pixel values (Figure 5).

Aside from this preprocessing, our model is very similar to the model used by Bellemare et al. (2014) and Veness et al. (2015). The CTS density model treats $x \in \mathcal{X}$ as a factored state, where each $(i, j)$ pixel corresponds to a factor $x^{i,j}$. The parents of this factor are its upper-left neighbours, i.e. pixels $(i - 1, j)$, $(i, j - 1)$, $(i - 1, j - 1)$ and $(i + 1, j - 1)$ (in this order). The probability of $x$ is then the product of the probability assigned to its factors. Each factor is modelled using a location-dependent CTS model, which predicts the pixel's colour value conditional on some, all, or possibly none, of the pixel's parents (Figure 6).

Original Frame (160x210)    Downsampled,
                            3-bit Greyscale (42x42)

Figure 5: Sample preprocessed image provided to the CTS model (**right**), along with the original frame (**left**). Although details are lost, objects can still be made out.

Figure 6: Depiction of the CTS "filter". Each downsampled pixel is predicted by a location-specific model which can condition on the pixel's immediate neighbours (in blue).

## C.2 A Taxonomy of Exploration

We provide in Table 1 a rough taxonomy of the Atari 2600 games available through the ALE in terms of the difficulty of exploration.

We first divided the games into two groups: those for which local exploration (e.g. $\epsilon$-greedy) is sufficient to achieve a high scoring policy (*easy*), and those for which it is not (*hard*). For example, SPACE INVADERS versus PITFALL!. We further divided the *easy* group based on whether an $\epsilon$-greedy scheme finds a *score exploit*, that is maximizes the score without achieving the game's stated objective. For example, KUNG-FU MASTER versus BOXING. While this distinction is not directly used here, score exploits lead to behaviours which are optimal from an ALE perspective but uninteresting to humans. We divide the games in the *hard* category into dense reward games (MS. PAC-MAN) and sparse reward games (MONTEZUMA'S REVENGE).

| Easy Exploration | | | Hard Exploration | |
|---|---|---|---|---|
| Human-Optimal | | Score Exploit | Dense Reward | Sparse Reward |
| ASSAULT | ASTERIX | BEAM RIDER | ALIEN | FREEWAY |
| ASTEROIDS | ATLANTIS | KANGAROO | AMIDAR | GRAVITAR |
| BATTLE ZONE | BERZERK | KRULL | BANK HEIST | MONTEZUMA'S REVENGE |
| BOWLING | BOXING | KUNG-FU MASTER | FROSTBITE | PITFALL! |
| BREAKOUT | CENTIPEDE | ROAD RUNNER | H.E.R.O. | PRIVATE EYE |
| CHOPPER CMD | CRAZY CLIMBER | SEAQUEST | MS. PAC-MAN | SOLARIS |
| DEFENDER | DEMON ATTACK | UP N DOWN | Q*BERT | VENTURE |
| DOUBLE DUNK | ENDURO | TUTANKHAM | SURROUND | |
| FISHING DERBY | GOPHER | | WIZARD OF WOR | |
| ICE HOCKEY | JAMES BOND | | ZAXXON | |
| NAME THIS GAME | PHOENIX | | | |
| PONG | RIVER RAID | | | |
| ROBOTANK | SKIING | | | |
| SPACE INVADERS | STARGUNNER | | | |

Table 1: A rough taxonomy of Atari 2600 games according to their exploration difficulty.

| | | | 0 | | 2 | | | |
|---|---|---|---|---|---|---|---|---|
| | 3 | 4 | 5 | 6 | 7 | | | |
| | 8 | 9 | 10 | 11 | 12 | 13 | 14 | |
| | 16 | 17 | 18 | 19 | 20 | 21 | 22 | 23 |

Figure 7: Layout of levels in MONTEZUMA'S REVENGE, with rooms numbered from 0 to 23. The agent begins in room 1 and completes the level upon reaching room 15 (depicted).

## C.3 Exploration in MONTEZUMA'S REVENGE

MONTEZUMA'S REVENGE is divided into three levels, each composed of 24 rooms arranged in a pyramidal shape (Figure 7). As discussed above, each room poses a number of challenges: to escape the very first room, the agent must climb ladders, dodge a creature, pick up a key, then backtrack to open one of two doors. The number of rooms reached by an agent is therefore a good measure of its ability. By accessing the game RAM, we recorded the location of the agent at each step during the course of training.[3] We computed the visit count to each room, averaged over epochs each lasting one million frames. From this information we constructed a map of the agent's "known world", that is, all rooms visited at least once. The agent's current room number ranges from 0 to 23 (Figure 7) and is stored at RAM location 0x83. Figure 8 shows the set of rooms explored by our DQN agents at different points during training.

Figure 8 paints a clear picture: after 50 million frames, the agent using exploration bonuses has seen a total of 15 rooms, while the no-bonus agent has seen two. At that point in time, our agent achieves an average score of **2461**; by 100 million frames, this figure stands at **3439**, higher than anything previously reported. We believe the success of our method in this game is a strong indicator of the usefulness of pseudo-counts for exploration.

We remark that without mixing in the Monte-Carlo return, our bonus-based agent still explores significantly more than the no-bonus agent. However, the deep network seems unable to maintain a sufficiently good approximation to the value function, and performance quickly deteriorates. Comparable results using the A3C method provide another example of the practical importance of eligibility traces and return-based methods in reinforcement learning.

## C.4 Improving Exploration for Actor-Critic Methods

Our implementation of A3C was along the lines mentioned in Mnih et al. (2016) and uses 16 threads. Each thread corresponds to an actor learner and maintains a copy of the density model. All the threads are synchronized with the master thread at regular intervals of 250,000 steps. We followed the same training procedure as that reported in the A3C paper with the following additional steps: We update our density model with the states generated by following the policy. During the policy gradient step, we compute the intrinsic rewards by querying the density model and add it to the extrinsic rewards before clipping them in the range $[-1, 1]$ as was done in the A3C paper. This resulted in minimal overhead in computation costs and the memory footprint was manageable ($<$ 32 GB) for most of the Atari games. Our training times were almost the same as the ones reported in the A3C paper. We picked $\beta = 0.01$ after performing a short parameter sweep over the training games. The choice of training games is the same as mentioned in the A3C paper.

The games on which DQN achieves a score of 150% or less of the random score are: ASTEROIDS, DOUBLE DUNK, GRAVITAR, ICE HOCKEY, MONTEZUMA'S REVENGE, PITFALL!, SKIING, SURROUND, TENNIS, TIME PILOT.

The games on which A3C achieves a score of 150% or less of the random score are: BATTLE ZONE, BOWLING, ENDURO, FREEWAY, GRAVITAR, KANGAROO, PITFALL!, ROBOTANK, SKIING, SOLARIS, SURROUND, TENNIS, TIME PILOT, VENTURE.

Figure 8: "Known world" of a DQN agent trained over time, with (**bottom**) and without (**top**) count-based exploration bonuses, in MONTEZUMA'S REVENGE.

The games on which A3C+ achieves a score of 150% or less of the random score are: DOUBLE DUNK, GRAVITAR, ICE HOCKEY, PITFALL!, SKIING, SOLARIS, SURROUND, TENNIS, TIME PILOT, VENTURE.

Our experiments involved the stochastic version of the Arcade Learning Environment (ALE) without a terminal signal for life loss, which is now the default ALE setting. Briefly, the stochasticity is achieved by accepting the agent action at each frame with probability $1 - p$ and using the agents previous action during rejection. We used the ALE's default value of $p = 0.25$ as has been previously used in Bellemare et al. (2016). For comparison, Table 2 also reports the deterministic + life loss setting also used in the literature.

Anecdotally, we found that using the life loss signal, while helpful in achieving high scores in some games, is detrimental in MONTEZUMA'S REVENGE. Recall that the life loss signal was used by Mnih et al. (2015) to treat each of the agent' lives as a separate episode. For comparison, after 200 million frames A3C+ achieves the following average scores: 1) Stochastic + Life Loss: 142.50; 2) Deterministic + Life Loss: 273.70 3) Stochastic without Life Loss: 1127.05 4) Deterministic without Life Loss: 273.70. The maximum score achieved by 3) is 3600, in comparison to the maximum of 500 achieved by 1) and 3). This large discrepancy is not unsurprising when one considers that losing a life in MONTEZUMA'S REVENGE, and in fact in most games, is very different from restarting a new episode.

### C.5   Comparing Exploration Bonuses

In this section we compare the effect of using different exploration bonuses derived from our density model. We consider the following variants:

- no exploration bonus,
- $\hat{N}_n(x)^{-1/2}$, as per MBIE-EB (Strehl and Littman, 2008);
- $\hat{N}_n(x)^{-1}$, as per BEB (Kolter and Ng, 2009); and
- $PG_n(x)$, related to compression progress (Schmidhuber, 2008).

The exact form of these bonuses is analogous to (4). We compare these variants after 10, 50, 100, and 200 million frames of training, again in the A3C setup. To compare scores across 60 games, we use

Figure 9: Average A3C+ score (solid line) over 200 million training frames, for all Atari 2600 games, normalized relative to the A3C baseline. Dotted lines denote min/max over seeds, inter-quartile range is shaded, and the median is dashed.

Figure 10: Inter-algorithm score distribution for exploration bonus variants. For all methods the point $f(0) = 1$ is omitted for clarity. See text for details.

inter-algorithm score distributions (Bellemare et al., 2013). Inter-algorithm scores are normalized so that 0 corresponds to the worst score on a game, and 1, to the best. If $g \in \{1, \ldots, m\}$ is a game and $z_{g,a}$ the inter-algorithm score on $g$ for algorithm $a$, then the score distribution function is

$$f(x) := \frac{|\{g : z_{g,a} \geq x\}|}{m}.$$

The score distribution effectively depicts a kind of cumulative distribution, with a higher overall curve implying better scores across the gamut of Atari 2600 games. A higher curve at $x = 1$ implies top performance on more games; a higher curve at $x = 0$ indicates the algorithm does not perform poorly on many games. The scale parameter $\beta$ was optimized to $\beta = 0.01$ for each variant separately.

Figure 10 shows that, while prediction gain initially achieves strong performance, by 50 million frames all three algorithms perform equally well. By 200 million frames, the $\hat{N}^{-1/2}$ exploration bonus outperforms both prediction gain and no bonus. The prediction gain achieves a decent, but not top-performing score on all games. This matches our earlier argument that using prediction gain results in too little exploration. We hypothesize that the poor performance of the $\hat{N}^{-1}$ bonus stems from too abrupt a decay from a large to small intrinsic reward, although more experiments are needed. As a whole, these results show how using PG offers an advantage over the baseline A3C algorithm, which is furthered by using our count-based exploration bonus.

|  | Stochastic ALE | | | Deterministic ALE | | |
|---|---|---|---|---|---|---|
|  | A3C | A3C+ | DQN | A3C | A3C+ | DQN |
| ALIEN | **1968.40** | 1848.33 | 1802.08 | 1658.25 | **1945.66** | 1418.47 |
| AMIDAR | **1065.24** | 964.77 | 781.76 | **1034.15** | 861.14 | 654.40 |
| ASSAULT | **2660.55** | 2607.28 | 1246.83 | **2881.69** | 2584.40 | 1707.87 |
| ASTERIX | 7212.45 | **7262.77** | 3256.07 | **9546.96** | 7922.70 | 4062.55 |
| ASTEROIDS | **2680.72** | 2257.92 | 525.09 | **3946.22** | 2406.57 | 735.05 |
| ATLANTIS | **1752259.74** | 1733528.71 | 77670.03 | 1634837.98 | **1801392.35** | 281448.80 |
| BANK HEIST | **1071.89** | 991.96 | 419.50 | **1301.51** | 1182.89 | 315.93 |
| BATTLE ZONE | 3142.95 | 7428.99 | **16757.88** | 3393.84 | 7969.06 | **17927.46** |
| BEAM RIDER | **6129.51** | 5992.08 | 4653.24 | 7004.58 | 6723.89 | **7949.08** |
| BERZERK | 1203.09 | **1720.56** | 416.03 | 1233.47 | **1863.60** | 471.76 |
| BOWLING | 32.91 | **68.72** | 29.07 | 35.00 | **75.97** | 30.34 |
| BOXING | 4.48 | 13.82 | **66.13** | 3.07 | 15.75 | **80.17** |
| BREAKOUT | 322.04 | **323.21** | 85.82 | 432.42 | **473.93** | 259.40 |
| CENTIPEDE | 4488.43 | **5338.24** | 4698.76 | 5184.76 | **5442.94** | 1184.46 |
| CHOPPER COMMAND | 4377.91 | **5388.22** | 1927.50 | 3324.24 | **5088.17** | 1569.84 |
| CRAZY CLIMBER | **108896.28** | 104083.51 | 86126.17 | 111493.76 | **112885.03** | 102736.12 |
| DEFENDER | **42147.48** | 36377.60 | 4593.79 | 39388.08 | 38976.66 | 6225.82 |
| DEMON ATTACK | **26803.86** | 19589.95 | 4831.12 | 39293.17 | 30930.33 | 6183.58 |
| DOUBLE DUNK | **0.53** | -8.88 | -11.57 | **0.19** | -7.84 | -13.99 |
| ENDURO | 0.00 | **749.11** | 348.30 | 0.00 | **694.83** | 441.24 |
| FISHING DERBY | **30.42** | 29.46 | -27.83 | **32.00** | 31.11 | -8.68 |
| FREEWAY | 0.00 | 27.33 | **30.59** | 0.00 | **30.48** | 30.12 |
| FROSTBITE | 290.02 | 506.61 | **707.41** | 283.99 | 325.42 | **506.10** |
| GOPHER | 5724.01 | **5948.40** | 3946.13 | **6872.60** | 6611.28 | 4946.39 |
| GRAVITAR | 204.65 | **246.02** | 43.04 | 201.29 | **238.68** | 219.39 |
| H.E.R.O. | **32612.96** | 15077.42 | 12140.76 | **34880.51** | 15210.62 | 11419.16 |
| ICE HOCKEY | **-5.22** | -7.05 | -9.78 | **-5.13** | -6.45 | -10.34 |
| JAMES BOND | 424.11 | **1024.16** | 511.76 | 422.42 | **1001.19** | 465.76 |
| KANGAROO | 47.19 | **5475.73** | 4170.09 | 46.63 | 4883.53 | **5972.64** |
| KRULL | 7263.37 | **7587.58** | 5775.23 | 7603.84 | **8605.27** | 6140.24 |
| KUNG-FU MASTER | **26878.72** | 26593.67 | 15125.08 | **29369.90** | 28615.43 | 11187.13 |
| MONTEZUMA'S REVENGE | 0.06 | **142.50** | 0.02 | 0.17 | **273.70** | 0.00 |
| MS. PAC-MAN | 2163.43 | 2380.58 | **2480.39** | 2327.80 | **2401.04** | 2391.89 |
| NAME THIS GAME | 6202.67 | **6427.51** | 3631.90 | 6087.31 | **7021.30** | 6565.41 |
| PHOENIX | 12169.75 | **20300.72** | 3015.64 | 13893.06 | **23818.47** | 7835.20 |
| PITFALL | **-8.83** | -155.97 | -84.40 | **-6.98** | -259.09 | -86.85 |
| POOYAN | 3706.93 | **3943.37** | 2817.36 | 4198.61 | **4305.57** | 2992.56 |
| PONG | **18.21** | 17.33 | 15.10 | **20.84** | 20.75 | 19.17 |
| PRIVATE EYE | 94.87 | **100.00** | 69.53 | 97.36 | **99.32** | -12.86 |
| Q*BERT | 15007.55 | **15804.72** | 5259.18 | 19175.72 | **19257.55** | 7094.91 |
| RIVER RAID | **10559.82** | 10331.56 | 8934.68 | **11902.24** | 10712.54 | 2365.18 |
| ROAD RUNNER | 36933.62 | **49029.74** | 31613.83 | 41059.12 | **50645.74** | 24933.39 |
| ROBOTANK | 2.13 | 6.68 | **50.80** | 2.22 | 7.68 | **40.53** |
| SEAQUEST | 1680.84 | **2274.06** | 1180.70 | 1697.19 | 2015.55 | **3035.32** |
| SKIING | -23669.98 | **-20066.65** | -26402.39 | **-20958.97** | -22177.50 | -27972.63 |
| SOLARIS | 2156.96 | **2175.70** | 805.66 | 2102.13 | **2270.15** | 1752.72 |
| SPACE INVADERS | **1653.59** | 1466.01 | 1428.94 | **1741.27** | 1531.64 | 1101.43 |
| STAR GUNNER | **55221.64** | 52466.84 | 47557.16 | **59218.08** | 55233.43 | 40171.44 |
| SURROUND | -7.79 | **-6.99** | -8.77 | **-7.10** | -7.21 | -8.19 |
| TENNIS | **-12.44** | -20.49 | -12.98 | -16.18 | -23.06 | **-8.00** |
| TIME PILOT | 7417.08 | 3816.38 | 2808.92 | **9000.91** | 4103.00 | 4067.51 |
| TUTANKHAM | **250.03** | 132.67 | 70.84 | **273.66** | 112.14 | 75.21 |
| UP AND DOWN | **34362.80** | 8705.64 | 4139.20 | **44883.40** | 23106.24 | 5208.67 |
| VENTURE | 0.00 | 0.00 | **54.86** | **0.00** | **0.00** | **0.00** |
| VIDEO PINBALL | 53488.73 | 35515.91 | **55326.08** | 68287.63 | **97372.80** | 52995.08 |
| WIZARD OF WOR | **4402.10** | 3657.65 | 1231.23 | **4347.76** | 3355.09 | 378.70 |
| YAR'S REVENGE | **19039.24** | 12317.49 | 14236.94 | **20006.02** | 13398.73 | 15042.75 |
| ZAXXON | 121.35 | **7956.05** | 2333.52 | 152.11 | **7451.25** | 2481.40 |
| Times Best | 26 | 24 | 8 | 26 | 25 | 9 |

Table 2: Average score after 200 million training frames for A3C and A3C+ (with $\hat{N}_n^{-1/2}$ bonus), with a DQN baseline for comparison.