[Reviews · NeurIPS 2016]

Reviewer 1

Summary

Count based exploration bonus is incorporated in several RL algorithms. When the state space is large, generalization over neighboring states is required to get a meaningful count measure. This paper proposes a transformation from density estimation to a count measure. The transformation is simple and local, and the resulting pseudo-count enjoys several attractive properties. The approach is demonstrated by adding exploration to some Atari2600 games.

Qualitative Assessment

I find the proposed scheme to be both elegant and effective. The presentation is quite clear. Adding some details on the density estimation scheme used in the application should be useful.

Confidence in this Review

2-Confident (read it all; understood it all reasonably well)


Reviewer 2

Summary

This paper presents how we can extend the notion of state-action visitation counts to reinforcement learning domains with large state spaces, where the traditional notion of visitation count becomes uninformatively sparse to be useful. The main idea comes harnessing density estimation to derive pseudo-counts that can be seen as equivalent to counts. Asymptotic analysis is given to show that they are indeed equivalent. The effectiveness of the pseudo-count derived from density estimation (CTS) is demonstrated by extending two RL algorithms (DQN and A3C) to accomodate count-based exploration, and running on a range of games in ALE.

Qualitative Assessment

This is a very nice paper that could be useful for many (deep) RL algorithms with large state spaces. I have a few comments regarding the pseudo-count: (1) In ICML this year, a number of exploration algorithms were presented that used model uncertainty for exploration. It seems to me that using model uncertainty is simpler and more natural, rather than using an external, density model. This issue is briefly mentioned in future directions section, but how does pseudo-count exploration compare to these model uncertainty exploration methods? When is it advantageous to use pseudo-counts? (2) How does the density-based counting scheme to a really simple counting scheme, e.g. partition the screen into big chunks and use color indicators? (3) In eqn (5), N(x) -> N_a(x) since we want the bonus to be dependent on states *and* actions?

Confidence in this Review

2-Confident (read it all; understood it all reasonably well)


Reviewer 3

Summary

This paper proposes a new "exploration bonus" approach based on the notion of "pseudo count". It demonstrates several desirable properties of pseudo count in two computer games: FREEWAY and PITFALL (Section 4); analyzes the asymptotic relationship between pseudo count and count (Section 5); discusses the connection to intrinsic motivation and information gain (Section 6); and shows some experiment results of using pseudo count for exploration (Section 7).

Qualitative Assessment

This paper is very interesting in general. To the best of my knowledge, the notion of "pseudo count", how to use this notion for exploration, and its connection to information gain, are novel. The experiment results in Section 4 and 7 are convincing. Overall, I think this paper has met the standard of NIPS poster, though the theoretical part of the paper is a little bit weak. My major concern of the paper is that it is not well-written in general. In particular, the transitions between sections are not very smooth. Moreover, the title of the paper indicates that this paper will focus on the relationships between pseudo count and intrinsic motivation (information gain). However, when reading the paper, I feel that this is only the focus of Section 6. Other sections seem to have different (even unrelated) foci. Please polish the final version of the paper. Minor comments: 1) figures in Figure 1 and 2 are too small. 2) what is the difference between C_n(x,a) in Section 2.1 and R^+_n(x,a) in Equation 5? I think they are the same (up to a factor of \beta), right?

Confidence in this Review

2-Confident (read it all; understood it all reasonably well)


Reviewer 4

Summary

This paper proposes an exploration strategy for deep RL. In particular, the authors derive a 'pseudo-count' from a sequential density model of the state space in ALE and claim to make explicit a relationship between information gain, prediction gain, and their proposed pseudo-count quantity. The authors present breakthroughs on ALE, making significant progress in Montezuma's Revenge.

Qualitative Assessment

Exploration is a key problem in RL; the authors revisit old ideas and derive new insights to improve on the state of the art. This is a good paper, with strong experimental results, but I still would like some additional explanations: - The pseudo-count quantity is derived from the so-called recoding probability of a density model, which is shown to be consistent with the empirical count. Say you have a specific state element x, which corresponds to some pseudocount N(x). Updating the density model with this element would lead to an updated pseudocount N'(x), which is consistent with the empirical count. But what happens to N(x) if you update the density model with a different element y? I would guess that N(x) can go down (which is obviously not consistent with the empirical count). This seems to relate to Fig. 1 (right),in which the pseudo-count has up/downward spikes. - Somehow I have the feeling that the notion of pseudo-count is somewhat redundant to the whole story. It seems that the authors can derive prediction gain (PG) and immediately derive information gain (IG), with no additional data needed. IG is well-established as a notion of intrinsic motivation. My question is, why work with pseudo-count? It seems to be quite a loose bound on PG/IG, which makes me think of it to be inherently less stable. If the whole pseudo-count measure would be removed from this paper, we basically have: measure the size in update to the density model and use this as an intrinsic reward. - It seems to me that the authors' definition of IG is different from what is used in literature (e.g., Planning to Be Surprised by Sun2011). In particular, it seems that an agent can gain infinite information over time by following a predefined policy, without actually exploring. This seems invalid. If the authors' IG quantity is different from what is generally established in literature, how could the authors relate it to it? If the authors' IG is a new quantity, what is the added value of of the authors' contribution 'a relation between IG and pseudo-counts'? Minor comments: - Corollary 1 reads quite dense. - In Fig 1 (left) the pseudo-count goes up in periods without salient events, is this due to generalization across states?

Confidence in this Review

2-Confident (read it all; understood it all reasonably well)


Reviewer 5

Summary

This paper has multiple contributions: 1. an extension to counting-based exploration based on sequential density model; 2. a simple but effective modification to DQN; 3. impressive empirical results incl. progress towards solving Montezuma's Revenge; 4. a connection to intrinsic motivation (more comments on this topic in detailed review below)

Qualitative Assessment

Overall this paper is a good paper that should be accepted. In particular, the formulation of pseudo-count seems novel, and Montezuma's Revenge experiment demonstrated convincingly the method can be effectively applied. I have however 2 concerns that I'd like the authors to address/respond to: 1. As stressed in title and in line 206, the claimed main result is the relationship between pseudo-count and information gain (IG), which is a classic concept in intrinsic motivation literature. The stated relationship, though technically sound, doesn't seem to relate to existing intrinsic motivation literature. I'd be happy to be corrected but as far as I can tell, the term "Information gain" here is used to define a quantity that's both technically and philosophically different from the usual Information gain. Specifically IG in this paper is defined over information gain over a mixture sequential density models of state visitations whereas IG is usually defined as information gain of "model of the environment" [1,2]. Philosophically, IG is introduced to create a "knowledge-seeking agent ... to gain as much information about its environment as possible" [2] and this is a characteristic that the "IG" definition in this paper fundamentally lacks. The reason is that if the behavior policy changes, there will always be information gain of state density model because state visitation frequencies change as the policy changes yet this specific "IG" doesn't need to reveal anything new about the environment, hence making the "IG" used in this paper distinctly different from the classic notion. The connection between "IG", PG, and pseudo-count is an interesting one but I'm wondering if the authors can justify how it "connects to intrinsic motivation", and "unifies count-based exploration and intrinsic motivation"? 2. Can the authors discuss more why pseudo-count is preferable to PG? From figure 2 in appendix it seems PG performs very competitively without extra tweaking on selecting what function to use to transform it but the inverse pseudo-count performs poorly with the /sqrt transformation. I suspect that if similar PG can be tried with different transformations it's going to outperform the tweaked pseudo-count bonus. Proof of Theorem 2 also indicates that 1/pseudo-count >= e^PG - 1. Since 1/psudo-count is exponentially larger than PG, it'd not be surprising that PG is better-behaved than 1/pseudo-count. I'm wondering if the authors can clarify if pseudo-count has distinctive advantages over PG or they are more of less interchangeable? [1]: Sun, Yi, Faustino Gomez, and Jürgen Schmidhuber. "Planning to be surprised: Optimal bayesian exploration in dynamic environments." International Conference on Artificial General Intelligence. Springer Berlin Heidelberg, 2011. [2]: Orseau, Laurent, Tor Lattimore, and Marcus Hutter. "Universal knowledge-seeking agents for stochastic environments." International Conference on Algorithmic Learning Theory. Springer Berlin Heidelberg, 2013.

Confidence in this Review

2-Confident (read it all; understood it all reasonably well)


Reviewer 6

Summary

The paper describes the use of intrinsic motivation for exploration in reinforcement learning. The measure of uncertainty used here is a new quantity called pseudo-count. It is able to work on large state spaces which is also shown in the experiment section with several hard games. The authors also show tight relations to other intrinsic motivation measures such as Bayesian information gain.

Qualitative Assessment

It would benefit the paper to include more connections to the cited reference "Variational Information Maximisation for Intrinsically Motivated Reinforcement Learning" as they also consider mutual information or channel capacity to be some form of path-counting. Technical quality: Good experiments with reasonably complex environments. Novelty/originality: Novel form of measuring uncertainty through counts. Relations to similar methods incomplete (c.f. section A in "Variational Information Maximisation for Intrinsically Motivated Reinforcement Learning"). Potential impact or usefulness: Useful since it works with large state spaces. Clarity and presentation: Clear structure.

Confidence in this Review

2-Confident (read it all; understood it all reasonably well)